# ADAPTIVE TEXT AND FEATURE EMBEDDING FOR CONSISTENT STORY GENERATION

## ABSTRACT

Recent advancements in text-to-image (T2I) generation have significantly improved image quality and text alignment. However, generating multiple coherent images that maintain consistent character identities across diverse textual descriptions remains challenging. Existing methods face trade-offs between identity consistency and per-image text fidelity, often yielding uniform poses or failing to capture specific details, resulting in inconsistent performance. In this paper, we explore text embeddings of word and PAD tokens from the scene descriptions, and ambiguity of the identity description. We find the identity and irrelevant components from the text embeddings to amplify and suppress them, respectively. Additionally, we detect under-specified identity descriptions and reuse their features during generative process. Finally, we introduce a unified evaluation protocol, the Consistency Quality Score (CQS), integrating identity preservation and per-image text alignment into a single comprehensive metric. CQS explicitly captures performance imbalances, aligning evaluation closely with human perceptual preferences. Our framework achieves state-of-the-art performance, effectively resolving prior trade-offs and providing valuable insights into consistent image generation.

## 1 INTRODUCTION

In recent years, text-to-image (T2I) generation models have significantly advanced, producing high-quality visuals closely aligned with textual prompts. Building upon these successes, users have shown growing interests in *story generation*: creating a sequence of coherent images with consistent character identities. The emergence of recent studies (Dinkevich et al., 2025; Liu et al., 2025; Tewel et al., 2024; Wang et al., 2025) and models such as NanoBanana highlights the increasing importance and practical demand for story generation. This task is crucial for narrative-driven content creation, animation, and interactive storytelling.

A primary challenge highlighted by recent studies is the inherent trade-off between identity preservation and prompt fidelity. For instance, ConsiStory (Tewel et al., 2024) and StoryDiffusion (Zhou et al., 2024) produce images with consistent identities but their poses are fixed and the images frequently disregard per-image prompts. One-Prompt-One-Story (Liu et al., 2025) produces images better aligned with per-image prompts but they contain inconsistent identities and elements from preceding prompts. Furthermore, these methods predominantly rely on earlier-generation architectures, i.e., U-Net-based models (Podell et al., 2023). This limits their capability to precisely align generated images with detailed and lengthy textual descriptions to meet recent demands. CharaConsist (Wang et al., 2025) struggles to produce a consistent identity when the description is under-specified, e.g., a man.

In this paper, we explore text embeddings of word and PAD tokens from the scene descriptions, and ambiguity of the identity description. 1) The text embeddings are inherently entangled via the text encoder and we amplify and suppress the components for subject identity and unrelated concepts, respectively. This unified strategy effectively overcomes the trade-offs inherent in previous text embedding methods, which typically employ separate mechanisms for identity preservation and text alignment. Additionally, we leverage padding embeddings from <pad> tokens as additionally semantic containers to effectively inject desired semantic information, motivated by recent findings that padding embeddings serve as semantic registers in text-to-image generation (Toker et al., 2025). Notably, we discover that padding embeddings contain not only meaningful semantic infor-

mation but also substantial irrelevant dummy semantics concentrated along a dominant embedding direction. Exploiting this characteristic, we selectively utilize only a portion of the padding embeddings as semantic containers to prevent dummy information from dominating our embedding modification. Consequently, our approach maintains robust text alignment performance irrespective of prompt length. Finally, we provide an in-depth analysis demonstrating how our method meaningfully improves the quality and consistency of generated images. These analyses and insights not only enhance consistent generation tasks but also provide valuable knowledge with potential applicability to broader tasks involving text-conditioned generative models.

We further highlight a critical limitation of previous identity preservation methods relying on image feature sharing (Tewel et al., 2024; Wang et al., 2025; Zhou et al., 2024). If the identity description is sufficiently detailed or clearly specifies the appearance (e.g., "a zebra"), the backbone alone typically ensures identity consistency without additional preservation strategies. Identity preservation methods are primarily beneficial for highly ambiguous descriptions (e.g., "a man"), which allow diverse identity appearances and thus require explicit consistency enforcement. However, existing feature-sharing methods uniformly apply identity preservation regardless of such ambiguity, leading to inconsistent performance across descriptions of varying specificity. To address this limitation, we propose an adaptive image feature-sharing method that automatically evaluates the ambiguity of the textual description. When the description yields diverse appearances, our method selectively applies additional feature sharing to reinforce identity consistency. Crucially, we design our feature-sharing strategy specifically for DiT blocks (Peebles & Xie, 2023), circumventing positional encoding biases by selectively sharing residual features that implicitly encode identity information.

In this task, prior studies have typically evaluated two crucial attributes, identity preservation and per-image text alignment, separately. However, separate evaluation struggles to effectively capture performance imbalances between these attributes, as strong performance in only one attribute indicates an underlying limitation of the model. Motivated by these insights, we propose a unified evaluation protocol, termed Consistency Quality Score (CQS), integrating identity preservation and per-image text alignment into a single comprehensive measure that explicitly highlights such imbalances.

Our main contributions can be summarized as follows:

- We propose a novel text embedding modification method that selectively controls identity-related embedding components and leverages additional semantic containers, effectively overcoming the trade-off between identity consistency and per-image text alignment.

- We introduce an adaptive image feature-sharing method specifically designed for DiT blocks, which selectively applies identity-preserving residual feature sharing based on the ambiguity of textual descriptions, thereby enhancing identity consistency.

- We propose a unified evaluation protocol termed Consistency Quality Score (CQS), integrating identity preservation and per-image text alignment into a single comprehensive measure to explicitly capture performance imbalances overlooked by separate evaluation approaches.

## 2 RELATED WORKS

### 2.1 CONSISTENT IMAGE GENERATION

Consistent generation aims to control subject identity across images while keeping each image faithful to its prompt. Recent work spans a wide range of topics, from compositional diversity (Dinkevich et al., 2025) to hand-drawn story synthesis (Zheng & Cun, 2025), yet the field ultimately aims to achieve both identity preservation and per-image text alignment. A common strategy encourages identity by sharing cross-image features via attention (Tewel et al., 2024; **?**; Wang et al., 2025). While effective, relying on this mechanism alone often yields rigid poses and reduces sensitivity to per-image instructions. Other approaches train additional encoders or fine-tune the model itself (Li et al., 2024; Guo et al., 2024; Ye et al., 2023; Shen & Elhoseiny, 2025; Liu et al., 2024). In practice, they often show at least one of limitations: restricted domain or training-data coverage, or trade-off between identity preservation and per-image text alignment. More recent works adopt powerful DiT-based T2I generation backbones to improve subject consistency (Tan et al., 2024; Xiao et al.,

2025; Labs et al., 2025) for a given input image. However, results can remain highly dependent on the provided reference image, making substantial appearance changes or diverse poses difficult.

Another line of work manipulates text embeddings. One-Prompt-One-Story (Liu et al., 2025) uses single shared text embeddings to preserve identity across frames and control attributes by either amplifying or damping embeddings with SVD. However, the model makes it difficult to separate identity from per-image attributes and leads to inconsistency. Furthermore, the results have dependency in text length. In contrast, we suggests a training-free text-embedding modification method with an adaptive image feature sharing that jointly improves identity preservation and per-image text alignment, mitigating the usual trade-off.

## 2.2 METRIC FOR CONSISTENT IMAGE GENERATION

To evaluate consistent image generation, two criteria are central: identity preservation and per-image text alignment. Standard feature-similarity evaluation measures these criteria separately, using VQAScore (Lin et al., 2024) for per-image text alignment and DINO or DreamSim (Caron et al., 2021; Fu et al., 2023) for identity preservation. Evaluating the two scores separately can be misleading because a model may excel in one and fail in the other. We propose a unified score that aggregates both criteria and penalizes their imbalance, yielding a single number that summarizes consistent-generation performance.

## 3 METHODS

### 3.1 PROBLEM SETTING

A consistent generation framework takes as input an identity prompt $p_{\mathrm{id}}$, which describes the shared identity to be preserved, along with multiple per-image prompts $\{p_i\}_{i=1}^{k}$ for individual images. Previous work (Liu et al., 2025) employs a single prompt $p_{\mathrm{single}} = [p_{\mathrm{id}}, p_1, \ldots, p_k]$ and modifying embeddings within this single prompt as an effective strategy to preserve the identity across generated images [1]. We adopt this single-prompt embedding modification strategy and detail the formulation below.

Given a single prompt $p_{\mathrm{single}}$ of length $L_{\mathrm{single}}$, the text encoder $E(\cdot)$ outputs $e_{\mathrm{single}} = [e_{\mathrm{id}}, e_1, \ldots, e_k] \in \mathbb{R}^{L_{\mathrm{single}} \times d}$, where $d$ is the embedding dimension. Separately, we denote the embeddings for the identity prompt and each per-image prompt as:

$$e_{\mathrm{id}} \in \mathbb{R}^{L_{\mathrm{id}} \times d}, \quad e_i \in \mathbb{R}^{L_i \times d}, \quad i = 1, \ldots, k. \tag{1}$$

Note that these embeddings are explicitly extracted from the single embedding $e_{\mathrm{single}}$ by splitting embedding dimensions according to the corresponding token indices, rather than obtained separately from individual prompt encodings.

When generating a specific image corresponding to prompt $p_i$, we refer to this selected prompt as the *expression prompt* $\{p_{\mathrm{exp}}\}$ with its embedding $\{e_{\mathrm{exp}}\}$. The remaining prompts $\{p_j\}_{j \in \{1,\ldots,k\}\setminus\{i\}}$, which should not influence the current image, are termed *suppression prompts*, with embeddings denoted $\{e_{\mathrm{sup}_j}\} = \{e_j\}_{j \in \{1,\ldots,k\}\setminus\{i\}}$.

For example, Fig. 1a shows how express and suppress embeddings are created. To generate an image from a second prompt, "dressed in a raincoat", we use the embeddings $\left[e_{\mathrm{id}}, e_{\mathrm{sup}_1}, e_{\mathrm{exp}}, e_{\mathrm{sup}_2}, \ldots, e_{\mathrm{sup}_k}\right]$ to effectively express the desired attributes (e.g., the dog and its raincoat) while suppressing irrelevant information.

### 3.2 SELECTIVE TEXT EMBEDDINGS MODIFICATION

Under the single prompt setting, the embedding $e_{\mathrm{single}}$ inherently contains entangled semantic across its dimensions (Chefer et al., 2023; Rassin et al., 2023; Kim et al., 2025). This entanglement requires more fine-grained control over the text embedding to get proper generation results. To address this, we propose a method to selectively manipulate the embedding: amplifying only the essential features from the expression, while downscaling the specific elements for suppression.

---

[1] compared to generating each image separately from individual prompts $[p_{\mathrm{id}}, p_i]$ for $i \in [1, k]$

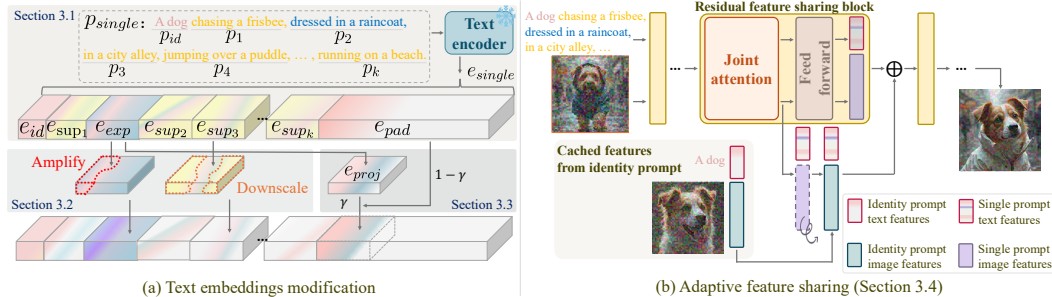

(a) Text embeddings modification

(b) Adaptive feature sharing (Section 3.4)

Figure 1: (a) shows brief illustration of our problem setting and text embeddings modification. (b) shows our adaptive feature sharing method with image features in Sec. 3.4

**Selective Expression.** To amplify embedding components for a desired expression, we enhance the components of the expression embedding that align with both an identity embedding and a expression embedding. Specifically, we treat the concatenation $[e_{\text{id}}, e_{\text{exp}}]$ as a reference expression embedding, denoted as $e_{\text{ref}-\text{exp}}$, and amplify only those components in $e_{\text{exp}}$ closely related to this reference.

This process begins by applying Singular Value Decomposition (SVD) separately on $e_{\text{exp}}$ and $e_{\text{ref}-\text{exp}}$. The SVD of $e_{\text{exp}}$ yields its singular vectors $\{v_i\}_{i=1}^{L_{\text{exp}}}$ and singular values $\{\sigma_i\}_{i=1}^{L_{\text{exp}}}$. Similarly, the SVD of $e_{\text{ref}-\text{exp}}$ produces its singular vectors $\{v_i\}_{i=1}^{L_{\text{ref}-\text{exp}}}$ and singular values $\{\sigma_i\}_{i=1}^{L_{\text{ref}-\text{exp}}}$, where $L_{\text{ref}-\text{exp}} = L_{\text{id}} + L_{\text{exp}}$.

We then form a reference vector $v_{\text{ref}-\text{exp}}$ as a weighted sum of singular vectors from $e_{\text{ref}-\text{exp}}$:

$$v_{\text{ref}-\text{exp}} = \sum_{i=1}^{L_{\text{ref}-\text{exp}}} \sigma_i v_i. \qquad (2)$$

This weighted sum emphasizes components dominant in both identity and expression attributes, serving as a meaningful reference for selective amplification.[2]

Next, we selectively amplify singular values $\sigma_i$ from $e_{\text{exp}}$ whose corresponding singular vectors $v_i$ exhibit cosine similarity above an adaptive threshold $\zeta_{\text{exp}}$ with the reference vector $v_{\text{ref}-\text{exp}}$:

$$\sigma_i; \leftarrow; U_{\text{exp}}(\sigma_i), \quad \text{if} \quad \cos(v_i, v_{\text{ref}-\text{exp}}) > \zeta_{\text{exp}}, \qquad \zeta_{\text{exp}} = \frac{1}{L_{\text{exp}}} \sum_{i=1}^{L_{\text{exp}}} \cos(v_i, v_{\text{ref}-\text{exp}}), \quad (3)$$

where the scaling function $U_{\text{exp}}$ is defined as $U_{\text{exp}}(\sigma) = \beta e^{\alpha\sigma}\sigma$.

**Selective suppression.** During suppression, our goal is to downscale embedding components associated with suppression prompts while preserving identity-related information. To achieve selective suppression, we focus on downscaling components in $e_{\text{sup}}$ unrelated to the identity prompt. Therefore, we use the identity embedding $e_{\text{id}}$ as a reference suppression embedding, denoted as $e_{\text{ref}-\text{sup}}$, to guide selective downscaling.

Similar to the expression process, we apply SVD to $e_{\text{sup}}$ and $e_{\text{ref}-\text{sup}}$ to obtain their respective singular vectors ($\{v_j^{\text{sup}}\}$ and $\{v_j^{\text{ref}-\text{sup}}\}$) and singular values ($\{\sigma_j^{\text{sup}}\}$ and $\{\sigma_j^{\text{ref}-\text{sup}}\}$). For this process, the reference suppression embedding's length is defined as $L_{\text{ref}-\text{sup}} = L_{\text{id}}$.

We then construct a reference vector $v_{\text{ref}-\text{sup}}$ using singular vectors and values from $e_{\text{ref}-\text{sup}}$, followed to Eqn. (2). This reference vector emphasizes components strongly related to the identity, serving as an anchor for selective suppression.

---

[2]To resolve the sign ambiguity of singular vectors, we align the reference orientation with the weighted mean of $e_{\text{exp}}$, a common convention in PCA/SVD Bro et al. (2008).

Finally, we selectively downscale singular values $\sigma_j$ from $e_{\sup}$ whose corresponding singular vectors $v_j$ have cosine similarity below an adaptive threshold $\zeta_{\sup}$ with the reference vector $v_{\mathrm{ref}-\sup}$:

$$\sigma_j; \leftarrow; D_{\sup}(\sigma_j), \quad \text{if} \quad \cos(v_j, v_{\mathrm{ref}-\sup}) < \zeta_{\sup}, \qquad \zeta_{\sup} = \frac{1}{L_{\sup}} \sum_{j=1}^{L_{\sup}} \cos(v_j, v_{\mathrm{ref}-\sup}), \quad (4)$$

where the downscaling function $D_{\sup}$ is defined as $D_{\sup}(\sigma) = \beta' e^{-\alpha' \sigma} \sigma$. This selective suppression effectively downscale unwanted suppression-prompt-related components while preserving critical identity semantics.

Finally, we reconstruct the text embeddings using the updated singular values, resulting in embeddings that effectively reflect the desired modifications. mjcAdditionally, this selective embedding modification also apply in text features in transformer block, before applied RoPE. Please refere the detail about n-th block/n-th step in appendix

### 3.3 IDENTITY SEMANTIC PROJECTION ON PADDING EMBEDDINGS

Recent studies indicate that these padding embeddings not only serve to maintain consistent embedding length but also can semantically influence generated images via joint attention mechanisms (Toker et al., 2025). Leveraging this semantic characteristic, we incorporate padding embeddings into our selective expression and suppression strategy.

First, we only use padding embeddings up to the length of the expression prompt, and concatenate this segment with the expression embedding $e_{\exp}$ for subsequent singular value decomposition (SVD). Using only a limited portion of padding embeddings is essential because, despite containing some valuable semantic information, padding embeddings predominantly carry irrelevant dummy information. Including excessive padding information can thus distort the intended expression or suppression semantics by overly emphasizing this irrelevant content (see Sec. 4.2 for further details).

Next, we directly project the expression embedding $e_{\exp}$ onto padding embeddings $e_{\mathrm{pad}}$ (or EOS embeddings $e_{\mathrm{eos}}$) to infuse them with meaningful semantics. Formally, this projection is defined as:

$$e_{\mathrm{proj}} = (P_{\exp}, e_{\mathrm{pad}}^\top)^\top, \qquad (5)$$

where the projection matrix $P_{\exp}$ is computed as:

$$P_{\exp} = e_{\exp}(e_{\exp}^\top e_{\exp})^\dagger e_{\exp}^\top, \qquad (6)$$

and $(\cdot)^\dagger$ denotes the pseudo-inverse. Finally, the updated padding embedding is calculated as:

$$e'_{\mathrm{pad}} = (1 - \gamma), e_{\mathrm{pad}} + \gamma, e_{\mathrm{proj}}. \qquad (7)$$

This adaptive semantic projection enhances text alignment by injecting expression semantics directly into padding embeddings, thereby effectively utilizing their semantic influence on image features.

### 3.4 ADAPTIVE FEATURE SHARING

When generating images from combinations of an identity prompt and per-image prompts $\{p_{\mathrm{id}}, p_1\}, \ldots, \{p_{\mathrm{id}}, p_k\}$, characters with detailed identity descriptions or with minimal appearance variance inherently maintain identity consistency with only selective text embeddings modification. However, prompts with high ambiguity require a supplementary identity preservation strategy. To this end, we propose *Adaptive Feature Sharing (AFS)*, a method that identifies high ambiguity of text and resolves identity shifts by adaptively sharing image features.

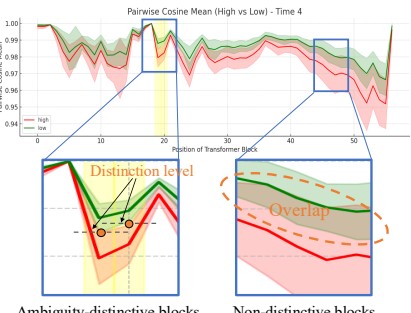

Figure 2: Comparison of cohesion distributions between ambiguity-distinctive and non-distinctive blocks

To determine whether an identity prompt is highly ambiguous, we first identify transformer blocks and denoising timesteps at which distinctions between high- and low-ambiguity cases become evident. Specifically, we locate intervals where the feature cohesion distributions of high- and low-ambiguity sample sets are clearly separable, exhibiting no overlap even when considering their standard deviations. As demonstrated in Fig. 2, these distinctions are particularly pronounced at transformer blocks 19 and 20 during denoising timesteps 3–6. For detailed criteria used in selecting this ambiguity threshold, please refer to Appx. A.

If a prompt set $\{p_{\text{id}}, p_1\}, \ldots, \{p_{\text{id}}, p_k\}$ is classified as highly ambiguous, we apply our image feature-sharing strategy to enhance identity consistency. Unlike traditional attention feature-sharing methods (Tewel et al., 2024; Zhou et al., 2024), which introduce positional biases within DiT blocks, we utilize residual feature sharing, which is inherently free from these biases. Specifically, we first generate and cache residual features conditioned on the only identity prompt $p_{\text{id}}$. During subsequent image generation using the single prompt, as illustrated in Fig. 1, we replace the residual features with the cached identity-conditioned features. This ensures robust identity consistency while preserving diversity in pose and scale.

## 4 ANALYSIS

### 4.1 REASON OF SELECTIVE TEXT EMBEDDINGS MODIFICATION

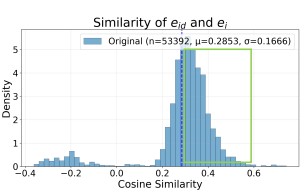
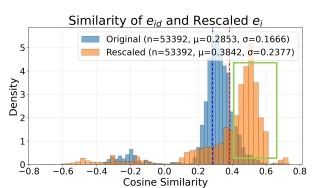
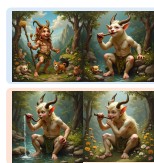

(a) Cosine similarity between $e_{id}$ and $e_i$

(b) selectively rescaled $e_i$ showing higher average similarity with $e_{id}$

(c) selectively scaling up improves identity similarity (**down**) then ampyfing all singular vectors of $e_i$ (**up**).

This section provides the background for the selective text embeddings modification introduced in Sec. 3.2. Our analysis indicates that the per-image prompt embedding contains not only the intended prompt semantics but also various other entangled attributes, including identity. As shown in Sec. 4.1a green box, positive similarity between per-image prompt embedding and identity embedding has positive similarity . Our selective expression which indicates the positive similarity in green box amplify expression-related semantic while preserving identity-related semantics.

Notably, embeddings also contain components negatively aligned with the identity embedding, rather than being simply orthogonal or neutral. Without selective modification (as in previous approaches (Liu et al., 2025)), uniformly amplifying all singular values in per-image prompt embeddings inadvertently increases the magnitude of negatively aligned components. This amplification of undesired semantics adversely impacts identity representation. Thus, selective amplification is crucial to prevent identity degradation, especially as scaling factors increase. For example, increasing scaling factors without selectivity fails to preserve identity consistency. This is because negatively aligned components specific to each prompt $p_i$ are amplified separately, disrupting shared identity semantics (see Sec. 4.1c - upper sample). In contrast, our selective expression method consistently amplifies embedding components commonly aligned across all $e_i$, thereby significantly improving identity consistency (see Sec. 4.1c - below sample).

Similarly, when per-image prompt embeddings are targeted for suppression, our selective suppression method explicitly preserves semantic components strongly associated with identity, ensuring that identity consistency remains unaffected.

### 4.2 DOMINANT INFORMATION OF PAD EMBEDDING

In this section, we provide empirical evidence explaining why only a portion of the padding embeddings should be used as a semantic containers, highlighting the dominance of irrelevant dummy information within the padding embeddings.

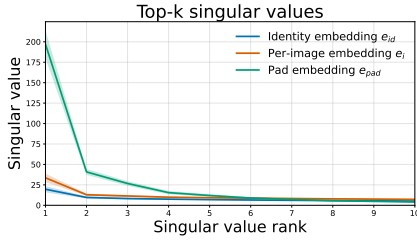 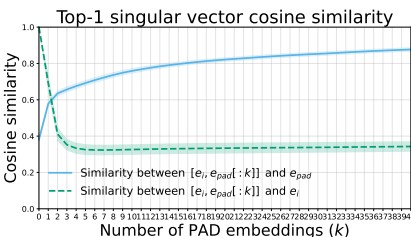

Figure 3: (a) Singular value magnitudes of $e_{id}$, $e_i$, and $e_{pad}$ (left). (b) Similarity of concatenated embeddings with the top-1 singular vectors of $e_{id}$ and $e_i$ as the number of concatenated Pad embeddings increases (right).

As shown in Sec. 4.2a, the singular values of the padding embedding $e_{pad}$ exhibit a significantly large magnitude at the top-1 singular value, with remaining singular values considerably smaller and similar in magnitude. This indicates that the padding embedding is predominantly aligned along a single direction. In contrast, singular values of the identity embedding $e_{id}$ and per-image prompt embedding $e_i$ exhibit relatively gradual magnitudes, indicating a more diverse semantic distribution.

We further demonstrate that the dominant singular direction in $e_{pad}$ represents dummy semantics rather than meaningful prompt-related information. In Sec. 4.2b, we show the similarity between a per-image embedding $e_i$ and its extended form $[e_i, e_{pad}[:n]]$, created by concatenating the first $n$ tokens of $e_{pad}$. As the length of padding embeddings concatenated to $e_i$ increases (i.e., as $n$ grows), the similarity to the original embedding $e_i$ significantly decreases, indicating that the additional padding embeddings introduce semantics unrelated to the original prompt. Conversely, similarity with $e_{pad}$ increases, further confirming that the dominant direction in padding embeddings carries prompt-irrelevant dummy information.

## 5 EXPERIMENTS

### 5.1 CONSISTENCY–QUALITY SCORE (CQS)

We propose a unified evaluation protocol, Consistency Quality Score (CQS), which combines identity preservation and per-image text alignment into a single measure, explicitly addressing performance imbalances overlooked by separate evaluations. Conventional evaluation reports VQA/CLIP score (text alignment/editability) and DINO/DreamSim (image similarity for identity consistency) as *separate* scores, which obscures the joint requirement that a model must *simultaneously* satisfy edit fidelity and identity preservation.

We therefore propose $CQS_{har}$, a single, balance-aware metric that combines VQA score with Dreamsim via a harmonic mean, yielding an interpretable summary of the trade-off.

Given a single prompt embedding $[e_{id}, e_1, \ldots, e_k]$, and corresponding images $X = \{x_1, \ldots, x_k\}$ where each $x_i$ is express $i$-th embedding $e_i$, while suppressing other embeddings. For each $i \in \{1, \ldots, k\}$ we define two VQA alignments:

$$t_i = VQA(x_i, [p_{id}, p_i]) \quad \text{and} \quad e_i = VQA(x_i, p_i), \tag{8}$$

corresponding to the frame prompt $(p_{id}, p_i)$ and the per-image prompt $p_i$, respectively. For identity consistency within $X$, let $D(x_i, x_j)$ be the dream similarity and define the per-image identity as

$$d_i = \frac{1}{k-1} \sum_{\substack{j=1 \\ j \neq i}}^{k} D(x_i, x_j). \tag{9}$$

Let $d_i$ denote the *identity score*[3] With the editing gap $\Delta_i = e_i - t_i$ and dataset-level means $\overline{\Delta}_+ = \mathbb{E}[\Delta_i \mid \Delta_i > 0]$ and $\overline{\Delta}_- = \mathbb{E}[\Delta_i \mid \Delta_i < 0]$, define penalty and reward score as:

$$\delta_i^- = (1-\lambda)|\overline{\Delta}_-| + \lambda \max(-\Delta_i, 0), \qquad \delta_i^+ = (1-\lambda)\overline{\Delta}_+ + \lambda \max(\Delta_i, 0). \tag{10}$$

---

[3]For DreamSim, we apply $1-\cdot$ and min–max scaling to match the vqa range $[VQA_{\min}, VQA_{\max}]$, ensuring equal weighting in the harmonic mean.

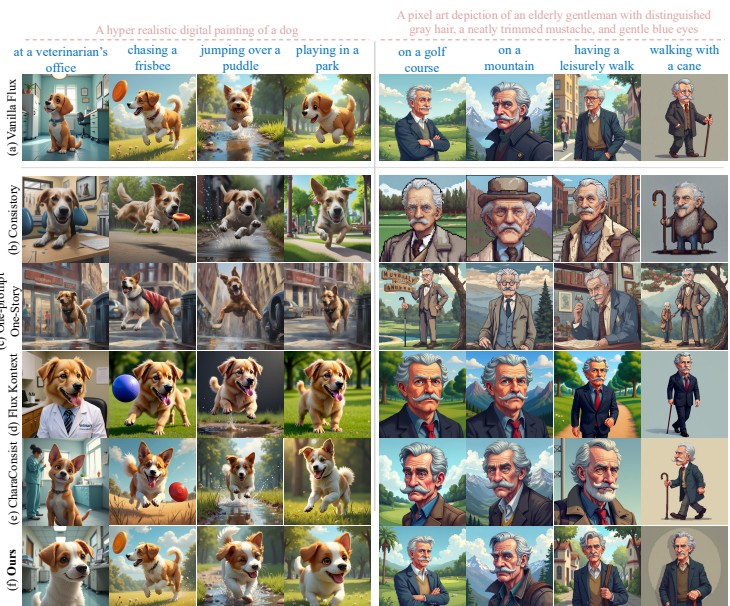

Figure 4: Qualitative results compared to other competitors.

The adjusted identity is then

$$d_i^* = d_i - \mu \, \mathbf{1}[\Delta_i < 0] \, \delta_i^- + \tau \, \mathbf{1}[\Delta_i > 0] \, \delta_i^+, \tag{11}$$

where $\lambda \in [0, 1]$ is a mixing weight and $\mu, \tau \geq 0$ are penalty/reward coefficients.[4] The per-image harmonic score is $h_i = \frac{2 \, e_i \, d_i^*}{e_i + d_i^* + \varepsilon}$ with a small $\varepsilon > 0$ for numerical stability. The overall $CQS_{har}$ computed as $\frac{1}{N} \sum_{i=1}^{N} h_i$. $CQS_{har}$ is a single-score measure of edit fidelity and identity preservation that proportionally down-weights samples whose per-image prompt alignment underperforms the total prompt.

|  | CQS$_{\mathrm{har}}$ ↑ | Per-image text alignment ↑ | Identity consistency ↓ |
|---|---|---|---|
| FLUX | 0.5680 | 0.6600 | 0.2616 |
| FLUX-kontext | 0.5684 | 0.6394 | 0.2165 |
| Consistory | 0.5829 | **0.6609** | 0.2402 |
| 1prompt1story | 0.5909 | 0.5873 | **0.1524** |
| CharaConsist | 0.5972 | 0.6477 | 0.2848 |
| **Ours** | **0.6100** | 0.6527 | 0.1885 |

Table 1: **Quantitative results.** Comparison with competing methods.

## 5.2 COMPARISON

As presented in Tbl. 1, our method achieves state-of-the-art performance in terms of the proposed $CQS_{har}$, which simultaneously considers identity consistency, text alignment, and performance imbalance. We argue that balanced performance across both identity consistency and text alignment aligns more closely with the objectives of consistent generation tasks, rather than excelling exclusively in one metric. For instance, the method One-Prompt-One-Story shows significantly poor identity consistency despite strong text alignment, resulting in lower overall performance when evaluated by $CQS_{har}$. Specifically, One-Prompt-One-Story ranks first in text alignment but drops to third in $CQS_{har}$ due to severe imbalance.

Similarly, ConsiStory achieves the highest text alignment score (though only slightly higher than ours), but its identity consistency, measured by DreamSim (Fu et al., 2023), is markedly high. This

---

[4]We fix the penalty $\mu$ and reward $\tau$ weights to 1 to ensure equal contribution, and use $\lambda = 1$ by default for fair comparison.

discrepancy arises from ConsiStory excessively sacrificing identity preservation to maximize text alignment. Since $CQS_{har}$ is the harmonic mean of DreamSim and the per-image and total text alignment score difference (penalty/reward), it penalizes such imbalance. Our method maintains strong text alignment and identity consistency simultaneously and thus delivers the best balanced performance.

Qualitative examples provided in Fig. 4 further illustrate our advantages, especially in scenarios with highly ambiguous identity descriptions (e.g., "a dog"). When descriptions contain detailed characteristics (e.g., "gentleman, gray hair, trimmed mustache, blue eyes"), even naive diffusion models such as FLUX inherently ensure good identity consistency. However, for ambiguous descriptions, identity consistency is significantly challenging for naive diffusion models. Our method effectively captures diverse per-image prompts (e.g., variations in poses and object scales) while maintaining strong identity consistency. This success is attributed to our adaptive feature sharing strategy, which implicitly shares identity-related features without introducing pose or scale biases. Conversely, methods like CharaConsist, which also rely on image feature sharing, fail to maintain identity consistency in highly ambiguous cases due to their uniform approach regardless of ambiguity. Furthermore, methods that use reference images, such as FLUX Kontext, tend to be heavily biased towards the pose or scale of the reference, often resulting in unnatural generations that do not accurately reflect the provided textual conditions (e.g., the first sample in the FLUX Kontext dog example). These limitations highlight the importance and effectiveness of our approach, which employs solely textual prompt sets for story generation, maintaining both flexibility and consistency.

### 5.3 ABLATION STUDY

| | CQS$_{har}$ ↑ | Per-image text alignment ↑ | identity consistency ↓ |
|---|---|---|---|
| FLUX | 0.4870 | **0.6450** | 0.2747 |
| All-SV + FI | 0.5280 | 0.6240 | 0.2217 |
| Ours w/o projected pad | 0.5522 | 0.6247 | 0.2291 |
| **Ours** | **0.5537** | 0.6271 | **0.2180** |

Table 2: **Comparison of baselines and our methods.** We compare the FLUX baseline, All-SV + FI (amplification along all singular directions with feature injection), w/o projection pad (removing Pad embeddings and the projection), and our full method.

In this section, an ablation study clarifies the performance impact of selective text-embedding modification, feature injection, and the projected-Pad method. We compare four settings. FLUX is the multi-prompt baseline. All-SV + FI amplifies and down-scales all singular directions of the per-image embedding $e_i$ with feature injection. w/o Pad & Proj removes Pad embeddings and the projection step and applies amplification and down-scaling to $e_i$ only. Ours is the full method.

FLUX achieves strong text–prompt alignment but exhibits low DreamSim. While All-SV + FI amplifies along all directions, Ours employs selective amplification, yielding better identity consistency and text alignment. In addition, by directly projecting per-image embeddings into the Pad embedding space, our method further improves text alignment over Ours w/o projected pad. In conclusion, our method achieves the highest $CQS_{har}$ score, demonstrating a better balance between per-image text alignment and identity consistency.

## 6 CONCLUSION

We proposed novel methods addressing key challenges in story generation, particularly the trade-off between identity consistency and per-image text alignment. Our selective text embedding modification method effectively controls identity-related semantics and leverages padding embeddings as semantic containers, ensuring robust alignment independent of prompt length. Additionally, our adaptive image feature-sharing strategy selectively reinforces identity consistency based on textual ambiguity. Finally, we introduced the Consistency Quality Score (CQS), a unified evaluation metric explicitly capturing performance imbalances between identity and text alignment. Our methods significantly advance consistent story generation and offer valuable insights for text-conditioned generative models.

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

## A DISCRIMINATIVE CRITERION FOR HIGHLY AMBIGUOUS DESCRIPTIONS

As mentioned in Sec. 3.4, we define the ambiguity level of an identity description. A *high ambiguity* description implies that given an identity prompt $p_{id}$ combined with multiple per-image prompts $(p_1, ..., p_k)$, the generated images exhibit diverse character appearances using a naive diffusion model. Conversely, a *low ambiguity* description indicates that generated images consistently represent the character's appearance across the prompt set.

To identify the specific transformer blocks and timesteps where ambiguity can be discriminatively assessed, we first create distinct high and low ambiguity sample sets. We generated approximately 100 prompt sets $\{p_{id}, p_1\}, \ldots, \{p_{id}, p_k\}$ using a Large Language Model (ChatGPT). The instructions provided to the LLM were: "Generate subjects and per-image prompts randomly with varying levels of appearance description, ranging from consistent appearances (e.g., "a zebra") to highly detailed descriptions (e.g., "a 16-year-old girl with wavy chestnut hair, a slender frame, and soft brown eyes"). Subjects may include animals, people, objects, food, etc., ensuring no overlap with our evaluation set."

Using these generated prompt sets, we synthesized images with a naive diffusion model (FLUX) and measured the visual similarity across images within each set using DreamSim. We labeled the prompt sets with the lowest 30% similarity scores as the high ambiguity set and those with the highest 30% similarity scores as the low ambiguity set.

Next, we computed feature cohesion (pairwise similarity) across batches while denoising each prompt set $\{p_{id}, p_1\}, \ldots, \{p_{id}, p_k\}$. Generally, high ambiguity samples exhibited lower cohesion, while low ambiguity samples had higher cohesion. However, considerable variation (high standard deviation) within prompt sets made clear separation challenging. Thus, we selected regions with no overlap between sets, taking standard deviations into account rather than focusing solely on the score differences.

As illustrated in Fig. 2, ambiguity-distinctive blocks clearly differentiate high and low ambiguity sets, whereas non-distinctive blocks exhibit significant overlap. We identified transformer blocks 19 and 20 at timesteps 3–6 as ambiguity-distinctive. If the cohesion scores at these blocks fall below thresholds of 0.985 and 0.987 respectively, we classify the identity prompt $p_{id}$ as highly ambiguous. If more than half of these blocks indicate high ambiguity, we confirm the prompt as highly ambiguous.

