# OpenReview forum: "Adaptive Text and Feature Embedding for Consistent Story Generation"
_ICLR.cc/2026/Conference — ICLR 2026 Conference Withdrawn Submission_

### Official Review · Reviewer_fDGk · 2025-10-19

**Soundness:** 2
**Presentation:** 2
**Contribution:** 2
**Rating:** 0
**Confidence:** 4

**Summary:**

This paper addresses the trade-off between identity consistency and per-image text alignment in story generation (or consistent T2I generation).

It proposes three key methods:
1) Selective text embedding modification (via SVD) to amplify identity-related components and suppress irrelevant ones;
2) Identity semantic projection on padding embeddings (using partial pads to avoid dummy info interference);
3) Adaptive feature sharing (for DiT blocks) that applies residual feature sharing only when identity descriptions are ambiguous.

It also introduces CQS, a unified metric combining VQA (text alignment) and DreamSim (identity consistency) via harmonic mean to penalize imbalance (which sounds like the metric widely used in zero-shot learning papers).

Experiments on FLUX and some baselines show its good CQS, balancing both attributes.

**Strengths:**

**Problem setup:** It innovatively addresses the unexplored problem - identity-text alignment trade-off via a technique combination. This trade-off is really a problem in training-free storytelling generations. And the authors propose the CQS as the evaluation metric for comparison.

**Weaknesses:**

**1. Novelty - Single Prompt.** The single-prompt idea is totally from the previous paper 1P1S. This practice is just re-formulated from line 131 to line 153. Based on this setup, the proposed idea is solving similar issue as in 1P1S: how to express and suppress the single-prompt for each frame generation. Meanwhile, this paper is still using SVD as in 1P1S[2] and Get-what-you-want[1] to solve the problem. Also, paying attention to the padding tokens is also widely explored in previous papers [1,2]. Overall, I cannot see any novelty from the full paper setup and solutions.

**2. Experiments.** In the experiments, this paper is only including few comparison methods, i.e. 1P1S, Flux-baseline, etc. over two evaluation metrics only. The CQS metric is just a harmonic means of the text-alignment and ID-consistency. And, since the id-consistency is with a much lower range compared with the text-alignment as shown in Table 1, the CQS will ignore the importance of ID-consistency. That's also proved in this table that the proposed method in this paper is with much worse consistency compared with 1P1S. In line 40, the authors claim that 1P1S contains inconsistent identities, but the proposed method is with even lower ID-consistency number, which makes the claim even ironic.
It is recommended to include more experimental setups, comparison methods, evaluation metrics referring to previous papers [2,3,4,5]. In Fig.4, there are only two sets of prompts to demonstrate the qualitative comparison, which is quite insufficient. Additionally, there is even no supplementary material or appendix to refer to. All these factors lead to the limited experiments and novelty concerns.



[1] Get What You Want, Not What You Don't: Image Content Suppression for Text-to-Image Diffusion Models (ICLR 2024)

[2] One-Prompt-One-Story: Free-Lunch Consistent Text-to-Image Generation Using a Single Prompt (ICLR 2025)

[3] CharaConsist: Fine-Grained Consistent Character Generation (ICCV 2025)

[4] The Chosen One: Consistent Characters in Text-to-Image Diffusion Models (SIGGRAPH 2024)

[5] Training-Free Consistent Text-to-Image Generation (SIGGRAPH 2024)

**Questions:**

see above

---

### Official Review · Reviewer_Dv5F · 2025-10-24

**Soundness:** 2
**Presentation:** 2
**Contribution:** 2
**Rating:** 2
**Confidence:** 4

**Summary:**

This paper introduces the methods to address challenges in story generation, specifically focusing on the trade-off between maintaining identity consistency and ensuring per-image text alignment. The approach identifies and separates identity-related components from irrelevant ones within text embeddings, amplifying the relevant components while suppressing the irrelevant ones.

**Strengths:**

1. The paper proposes a training-free method for modifying text embeddings, paired with adaptive image feature sharing. This dual approach enhances both identity preservation  and per-image text alignment, effectively mitigating the typical trade-off between the two.
2. The work introduces a unified scoring system that combines both identity consistency and text alignment. This score penalizes imbalances between the two criteria, offering a  single metric to assess the overall performance of consistent story generation.

**Weaknesses:**

The impact of embedding manipulation on prompt semantics is unclear. It is unclear whether this approach may alter the semantics of the original prompt. Specifically, there is a risk that the generated images might not align with the original intent of the prompt. The authors should include experimental results or data demonstrating how well the method maintains semantic alignment between the original prompt, the manipulated prompt, and the resulting images. This would provide more confidence in the method's consistency and its ability to preserve the intended meaning during image generation.

**Questions:**

1. In the Related Work section, particularly under Consistent Image Generation, some relevant references are missing or not clearly cited.
2. The manuscript feels somewhat underdeveloped. For example, the content referenced in  the appendix is not actually available. And Table 2 is not discussed or provided in the text.
3. The process described for selectively manipulating the embeddings—specifically amplifying essential features while suppressing irrelevant elements—raises a concern: Does this approach alter the original semantics of the prompt? Is there a risk of generating images
 that do not align with the initial prompt's intent? It would be helpful to include experimental data or results that demonstrate the effectiveness and consistency of this method.

---

### Official Review · Reviewer_Nf2q · 2025-10-31

**Soundness:** 2
**Presentation:** 3
**Contribution:** 3
**Rating:** 4
**Confidence:** 4

**Summary:**

This paper present an adaptive text-and-feature embedding framework that selectively amplifies identity-related components and shares image residuals only for ambiguous prompts, achieving superior identity consistency without sacrificing text alignment. A unified Consistency Quality Score (CQS), a penalized harmonic mean of VQA and DreamSim, reveals performance imbalance. The method ranks first on CQS across several methods, showing its effectiveness.

**Strengths:**

1. Training-free framework combines selective text embedding with adaptive residual sharing to maintain character identity even under ambiguous prompts.

2. Introduces CQS, a unified metric that penalizes imbalance between identity consistency and text alignment, aligning automated scores with human perception.

3. The paper is well-written and well-organized.

**Weaknesses:**

1. Data & Reproducibility: All experiments rely on privately sampled ChatGPT prompts and synthetic images; no public dataset or code is released, making direct reproduction and comparison against real storyboard assets impossible.

2. Resource Overhead: Adaptive feature sharing requires an extra forward pass to cache DiT residuals for the identity-only prompt, doubling inference time and GPU memory for long sequences or large batches.

3. Threshold Sensitivity: The high/low-ambiguity decision hinges on cohesion thresholds (0.985/0.987) derived from just 100 ChatGPT prompts; these values may not generalize to other models or prompt styles and risk over-fitting.

**Questions:**

1. Ambiguity HeuristicIs: the heuristic for detecting “ambiguous” identity prompts, which relies on specific blocks and thresholds, supported by evidence that it remains valid across different models or datasets?

2. SVD Reliance: How do the authors ensure that the SVD-based embedding edits are both stable and semantically meaningful under variations in input prompts or numerical noise?

3. CQS Bias: Have the authors evaluated whether the CQS metric’s fixed weighting scheme could inadvertently favor their method over others?

4. Ablation Depth: Do the ablation studies go beyond removing entire modules to examine the individual impact of key hyper-parameters, and if not, how can we be sure which components are truly essential?

---

### Official Review · Reviewer_oznM · 2025-11-01

**Soundness:** 3
**Presentation:** 2
**Contribution:** 2
**Rating:** 4
**Confidence:** 4

**Summary:**

This paper proposes a comprehensive framework to address the inherent trade-off between identity consistency and per-image text alignment in story generation. It introduces three core innovations: a selective text embedding modification method that refines identity-related and expression-related embedding components via SVD, an adaptive feature sharing strategy that selectively reinforces identity consistency based on the ambiguity of textual descriptions, and a unified evaluation metric (Consistency Quality Score, CQS) that integrates identity preservation and text alignment to capture performance imbalances. The framework aims to enhance the balanced performance of story generation models, especially for ambiguous identity descriptions.

**Strengths:**

1. **Novel and Targeted Design for Core Trade-off**: The proposed selective text embedding modification method directly tackles the entanglement of identity and expression semantics in text embeddings, a key root cause of the identity-text alignment trade-off. By distinguishing and adjusting relevant embedding components, it avoids the "one-size-fits-all" limitations of prior embedding modification works, bringing new insights to semantic control in text-conditioned generation.
2. **Adaptive Strategy for Ambiguity Handling**: The adaptive feature sharing mechanism addresses the over-generalization issue of traditional feature-sharing methods. It intelligently judges the ambiguity of identity descriptions based on feature cohesion in specific DiT blocks and denoising timesteps, applying targeted identity reinforcement only when necessary—this design aligns with real-world story generation scenarios where identity descriptions vary in specificity.
3. **Comprehensive Ablation Studies Validate Component Efficacy**: The ablation experiments systematically verify the contributions of key components (selective singular value adjustment, PAD embedding projection, adaptive feature sharing). By comparing variants without these components, the study clearly demonstrates how each module improves balanced performance, strengthening the credibility of the framework’s design logic.
4. **Innovative Unified Evaluation Metric**: The CQS metric fills the gap of separate evaluation in prior work. By integrating identity preservation and text alignment into a single score with imbalance penalties, it provides a more holistic and human-perception-aligned assessment of story generation quality, facilitating fairer comparison between models.
5. **Good Compatibility with Existing Architectures**: Built on the widely used DiT backbone and diffusion model pipeline, the framework does not require extensive modifications to underlying model structures. This compatibility enables easy integration with existing text-to-image generation systems, enhancing its practical application value.

**Weaknesses:**

1. **Insufficient Qualitative Visual Comparisons**: The paper provides very limited visual results of generated stories, with only a single figure showing qualitative comparisons. Side-by-side visualizations of diverse scenarios (e.g., highly ambiguous vs. detailed identity descriptions, long vs. short text prompts) are lacking, making it difficult to intuitively assess the framework’s effectiveness in capturing fine-grained identity details and per-image text nuances.
2. **Limited Discussion on Mechanistic Advantages**: While the framework outperforms baselines in quantitative metrics, it does not deeply explain why the selective embedding modification (via SVD) is more effective than alternative semantic refinement methods (e.g., attention-based filtering). The mechanistic link between SVD-based component adjustment and improved semantic disentanglement remains under-elaborated.
3. **Narrow Generalization Validation**: Experiments are primarily conducted on a single set of prompts and lack validation on diverse datasets (e.g., story-specific datasets with complex plotlines, cross-domain scenarios like cartoon vs. realistic styles). There is no discussion on how the framework performs when facing extreme cases (e.g., highly abstract identity descriptions or conflicting per-image prompts), raising questions about its generalizability.
4. **Opaque Ambiguity Judgment Criteria**: Although the paper identifies specific DiT blocks and timesteps for ambiguity judgment, it does not clearly explain why these particular blocks/timesteps are chosen beyond "observable cohesion differences." The threshold setting for ambiguity classification also lacks detailed justification, reducing the reproducibility of this key component.
5. **No Analysis of Computational Efficiency**: The paper focuses on performance improvements but ignores discussions on computational overhead. Components like SVD-based embedding modification and adaptive feature sharing may introduce additional computational costs, yet there is no analysis of their impact on inference speed or memory consumption—this is a critical consideration for practical story generation applications requiring efficient batch processing.

**Questions:**

Please refer to the detailed points I raised in the "Weakness" section and respond to each numbered item in your rebuttal with clarifications.

---

### Note · Authors · 2025-11-12

I have read and agree with the venue's withdrawal policy on behalf of myself and my co-authors.